# Enteral Nutrition Supplemented with Transforming Growth Factor-β, Colostrum, Probiotics, and Other Nutritional Compounds in the Treatment of Patients with Inflammatory Bowel Disease

**DOI:** 10.3390/nu12041048

**Published:** 2020-04-10

**Authors:** John K. Triantafillidis, Maria Tzouvala, Eleni Triantafyllidi

**Affiliations:** 1“Metropolitan General” Hospital, ZC 15562 Holargos, Greece; eltriant@yahoo.gr; 2Department of Gastroenterology “St Panteleimon” General Hospital, ZC 18454 Nicea, Greece; tzouvalam@gmail.com

**Keywords:** enteral nutrition, TGF-β, colostrum, probiotics, Crohn’s disease, ulcerative colitis, inflammatory bowel disease

## Abstract

Enteral nutrition seems to play a significant role in the treatment of both adults and children with active Crohn’s disease, and to a lesser degree in the treatment of patients with active ulcerative colitis. The inclusion of some special factors in the enteral nutrition formulas might increase the rate of the efficacy. Actually, enteral nutrition enriched in Transforming Growth Factor-β reduced the activity index and maintained remission in patients with Crohn’s disease. In addition, a number of experimental animal studies have shown that colostrum exerts a significantly positive result. Probiotics of a special type and a certain dosage could also reduce the inflammatory process in patients with active ulcerative colitis. Therefore, the addition of these factors in an enteral nutrition formula might increase its effectiveness. Although the use of these formulas is not supported by large clinical trials, it could be argued that their administration in selected cases as an exclusive diet or in combination with the drugs used in patients with inflammatory bowel disease could benefit the patient. In this review, the authors provide an update on the role of enteral nutrition, supplemented with Transforming Growth Factor-β, colostrum, and probiotics in patients with inflammatory bowel disease.

## 1. Introduction

Enteral nutrition (EN) can reduce Crohn’s disease (CD) activity and maintain remission in both children and adults [1,2]. According to the international guidelines, nutritional support using orally administered liquid formulas should be considered for CD patients, serious cases of ulcerative colitis (UC), in combination with steroids in undernourished individuals, in cases of steroid intolerance, and in patients with stenosis of the small intestine [1,2,3,4]. Transforming Growth Factor-β (TGFβ), a polypeptide present in both human and bovine milk, has been shown to play an important role in the development of immune tolerance and the prevention of autoimmunity, being concurrently an inhibitor of intestinal epithelial cell growth and a stimulator of intestinal epithelial cell differentiation [5].

Studies concerning the administration of EN formulas containing TGFβ in both adults and children with active and inactive CD have been previously published, although their promising results were based on a small number of patients [6,7,8]. Colostrum is the first milk produced after birth, being rich in a number of bioactive molecules, including growth factors. It has been experimentally used in rats, and the results of its administration were encouraging [9,10], although, again, there is a lack of controlled clinical studies. Finally, probiotics of a special type and dosage could in fact reduce the inflammatory process in patients with active ulcerative colitis UC [11,12,13].

During the last two decades, industry efforts resulted in the production of food containing TGFβ and other molecules that have biological activities for patients with inflammatory bowel disease IBD. The hope is that the addition of these biological molecules in the same EN formula might increase the beneficial effect expected from the administration of EN not containing these factors.

In this review the authors analyze the theoretical basis of adding TGFβ, colostrum, probiotics, and other bioactive compounds, including flavonoids, short chain fatty acids, and bioactive peptides, in an EN formula. Then, they try to critically evaluate the available data regarding the effectiveness of this formula in patients with IBD.

## 2. Tranforming Growth Factor-β

TGFβ is a cytokine produced by all white blood cell lineages, belonging to the TGF superfamily, which is a family of proteins named after the first member TGFβ1. It includes three different isoforms and other signaling proteins. It consists of ligands and receptors that all signal through downstream mediators, termed the Smads. Secreted TGFβ is a latent complex, proteolytically cleaved in the extracellular space resulting in the release of TGFβ dimmer allowing TGFβ to bind to cell surface receptors [14,15].

It is present in both human and bovine milk in high concentrations (20–40 mg/L and 1–2 mg/L respectively). The amount of TGFβ being present in milk could maintain the gastrointestinal integrity in suckling human neonates.

Active TGFβ regulates cell adhesion and apoptosis through the TGFβ signaling pathway. It also plays an important role in the development of immune tolerance and the prevention of autoimmunity, concurrently being an inhibitor of intestinal epithelial cell growth and a stimulator of intestinal epithelial cell differentiation. It is a potent chemo-attractant for neutrophils and stimulates epithelial cell migration at wound sites, thus facilitating wound repair. TGFβ is the master regulator to drive fibrosis and angiogenesis in the intestine [16]. TGFβ plays an important role in the function of a number of immune cells, including T and B lymphocytes, macrophages, as well as in the cell cycle [17]. Among its key functions is the regulation of inflammatory processes, particularly in the gut [18,19]. TGFβ also plays a crucial role in stem cell differentiation, as well as in T cell regulation and differentiation [20]. Neutralization of TGFβ in vitro inhibits the differentiation of helper T cells into T_h_17 cells.

TGFβ inhibits B cell proliferation via the induction of the transcription factor Id3, blocks B cell activation and promotes class switching Immunoglobulin A (IgA) in both human and mouse B cells. TGFβ, also, has an inhibitory function for antibody production [21] and induces the apoptosis of immature or resting B cells. It stimulates resting monocytes and inhibits activated macrophages, acting as a chemoattractant for monocytes [22] and downregulates inflammatory cytokine production by monocytes and macrophages through the inhibition of NF-κB [23].

TGFβ promotes IgA production as well as oral tolerance towards cow’s milk. Low levels of IgA have been observed in the colostrum and breast milk of mothers of offspring who experienced a cow’s milk allergy [24]. Therefore, TGFβ plays an important role in maintaining tolerance against self antigens and antigens derived from food, commensal bacteria, and fetal alloantigens [14]. It also plays a crucial role in the regulation of the cell cycle by blocking progress through suppressing the expression of c-myc, a gene which is involved in G1 cell cycle progression [25].

### TGFβ and Inflammatory Bowel Disease

TGFβ is involved in various chronic inflammatory disorders, including IBD. It has been shown long since that TGFβ and its receptors were increased in intestinal cells of patients with IBD, especially in CD patients [26], a fact confirmed in experimental animal models [16]. TGFβ functions by binding to two receptors, type I subtype ALK5 and type II, which are concomitantly required for signal transduction. TGFβ, also, binds to its type III receptor (betaglycan), creating a heteromeric complex with TGFβ type II receptor. Binding of the ligands to type II receptors activates the type I receptor. The activated TGFβ receptors phosphorylate Smad2 and Smad3 heterodimer subsequently interacting with Smad4. The Smad2/3-Smad4 complex translocates from the cytosol to the nucleus and binds to other DNA-binding co-factors regulating specific TGFβ target genes, including fibronectin and collagen [27]. The concomitant over-expression of TGFβ and their signaling receptors in CD means that these regulatory molecules might play a role in the pathophysiology of CD. The activation of TGFβ-mediated pathways might promote the repair of mucosal injury by enhancing the process of reepithelization, concurrently promoting, however, extracellular matrix generation, intramural fibrosis and intestinal obstruction [28]. Myofibroblasts derived from smooth muscle cells in chronic inflammatory situations, can increase fibrosis in IBD through the production of collagen and matrix metalloproteinases due to stimulation of TGF-β [29].

Antibodies inhibiting the binding of the ligand to the receptor and antisense oligonucleotides resulting in the reduction of the expression of TGFβ, such as Smad6 and Smad7, have been developed [30,31]. The inhibition of Smad7 using antisense oligonucleotides decreased inflammation in an animal model of TNBS colitis [32] and improved the clinical activity of patients with CD [33] The orally administered antisense oligonucleotide of Smad7 (mongersen) showed a clinical efficacy in CD patients, although high doses induced bowel obstruction caused by TGFβ signaling promoting fibrosis [34]. Another potent proinflammatory cytokine involved in TGF-β signaling is IFN-γ, a cytokine that inhibits the TGFβ-induced phosphorylation of Smad3 and the activation of TGFβ genes by inducing the expression of Smad7 [35].

The critical role of TGFβ in the development and homeostasis of intestinal immunity and the CNS in humans was recently stressed by Kotlarz et al., who described three individuals with biallelic loss-of-function mutations in the *TGFβ* gene presented with severe infantile IBD and CNS disease [36].

## 3. Colostrum in Health and Disease

Colostrum should be defined as the milk produced in the first 48 h after birth. It is rich in a wide range of antimicrobial peptides, immune-regulating components and growth factors harboring anti-inflammatory and immuno-modulatory properties. Human breast milk is rich in nutrients, hormones, growth factors and immunoactive molecules with anti-inflammatory and immunomodulatory properties, which can influence the growth, development and immune status of the infant [37]. The pivotal functions of colostrum are to provide essential nutritional components, reinforce natural defenses, modulate intestinal microflora and immune responses, and promote the growth, maturation and repair of many tissues [38].

During breast feeding, all the isoforms of the TGFβ are produced, the majority being TGFβ2 [39]. The levels of this cytokine range between 0.1 and 13.3 μg/L in term colostrum and between 1.4 and 43 μg/L in preterm colostrum. These levels decrease along the lactation period with concentrations of 0.4–2.8 μg/L in term and 0.9–6.3 μg/L in preterm mature milk [40].

The bioactive substances present in milk and colostrum that might be useful in a number of pathological situations, are shown in Table 1. It is important to bear in mind that many of the factors present in milk and colostrum, e.g., Epidermal Growth Factor (EGF), TGFα, TGFβ, amphiregulin, betacellulin, and heparin-binding EGF-like peptides, are also produced in the gut, explaining, at least in part, why colostrum promotes the repair of the gut mucosa [41]. Colostrum whey proteins have been administered at a dose of 30 g/d for 6 months in humans without any kind of side-effects [42].

Colostrum is rich in biologically active molecules that have specific functions in the physiology of the digestive system. It has been shown that colostrum induces epithelial regeneration via the stimulation of cellular proliferation and differentiation. Colostrum growth factors and lactoferrin could modulate the function of intestinal cells [43]. Orally administered bovine Lactoferrin inhibits vascular EGF-induced angiogenesis in normal rats [44]. Furthermore, colostrum inhibits the expression of inflammatory genes following invasion by enteric pathogens, reduces the inflammatory cytokines (IL-1β, IL-8, and TNFα) and inhibits the NF-κB pathway [45]. The oligosaccharides (fructo-oligosaccharides, beta-galactooligosaccharides, gangliosides, and nucleosides) present in bovine milk selectively stimulate the growth of beneficial bacteria acting as prebiotics [46]. It seems that colostrum is able to preserve the intestinal microbiota, which enforces the mucosal barrier, thus reducing the inflammatory reactions. Beneficial Gram negative bacteria can, also, reduce the cell surface expression level of the LPS receptor complex and, subsequently, the activation of the TLR4 signaling pathway. Based on these data, it might be argued that bovine colostrum may reduce the side-effects of drugs used in the treatment of IBD, although this assumption needs to be confirmed in clinical trials. Bovine colostrum contains high levels of immunoglobulins that can reduce antigenic stimulation derived from gut endotoxins (LPS), thus improving the state of tolerance and reducing the risk of bacterial translocation [47]. Finally, colostrum can prevent gastrointestinal infections in infants, as it has been noticed that infants on a formulated diet are more susceptible to infectious diarrhea compared to breast-feeding infants [48].

### 3.1. Clinical Studies of Colostrum in Patients with IBD

While several studies have evaluated the effects of colostrum administration in the prevention and treatment of various gastrointestinal disorders in different animal species and humans [49,50,51,52,53], studies concerning the clinical use of bovine colostrum in IBD are still limited and controversial [54]. In a small study, Khan et al. investigated the role of bovine colostrum in 14 patients with mild to moderately severe distal colitis. Patients received colostrum enemas (100 mL of 10% solution, eight patients) or placebo (albumin solution, six patients) twice daily for 4 weeks in conjunction with their regular treatment with mesalazine (1.6g/d). After 4 weeks, the colostrum group had a mean reduction in symptom score of −2.9, whereas in the placebo group the corresponding feature was +0.5. The histological score improved in five of eight patients in the colostrum group. This small encouraging study has not been, so far, repeated, although it is worth consideration [55].

### 3.2. Role of Colostrum in Ameliorating Chemical Colitis

Experimental data concerning the role of colostrums in ameliorating chemical colitis have been published during the last years, all of which confirmed the beneficial effect of colostrum and, at the same time, clarified the various mechanisms being responsible for the improvement of experimental colitis (Table 2).

Bodammer et al. showed that prophylactic administration of colostrum was able to improve clinical symptoms and colonic inflammation in a DSS model of colitis, concurrently redistributing immune-regulatory, peripheral and splenic gd TCR+ cells, and CD11b+Gr1+ cells [9]. In a subsequent experimental study, aiming to explore the underlying molecular mechanisms, the same group of investigators found an increased claudin-2 expression in the distal ileum of healthy mice after being fed with colostrum, whereas other tight junction proteins remained unchanged. Interestingly enough, the elevation of claudin-2 was accompanied by neither an increased ion permeability nor an impaired barrier function. In an in situ perfusion model, a 1h exposure of the colonic mucosa to colostrum induced increased MRNA levels of barrier-strengthening TGFβ, a fact that might have compensated for the claudin-2 increase and contributed to the barrier strengthening effects of colostrum [56].

Kailash et al. investigated two polyclonal antibodies (AVX-470 and AVX-470m) specific to human TNF isolated from the colostrum of dairy cows that had been previously immunized with TNF, in two models of experimental colitis (DSS and TNBS-induced colitis) [57]. These orally administered antibodies significantly reduced the severity of colitis in a manner comparable with that of oral prednisolone. The authors concluded that the AVX-470 polyclonal anti-TNF antibody has in vitro activity comparable to that of infliximab. One important question concerning this study was related to the absence of breaking-down the administered therapeutic antibodies in the hostile gastric and bowel environment as they were passing through the entire bowel. However, it was found that bovine antibodies from milk or colostrum are suitable for oral delivery by virtue of their stability of digestion in the gastrointestinal tract [58]. Other reports, also, have supported the use of orally administered antibodies from bovine colostrum for the treatment of various disorders [59]. The utilized antibodies were designed to act within the gut lumen against infectious agents. These studies established the safety and efficacy of bovine antibodies, and supported the use of bovine colostrum antibodies as gut-targeted therapeutics. In accordance with the abovementioned studies, the minimally absorbed peptide linaclotide has been approved for the treatment of irritable bowel syndrome [60]. Therefore, bovine colostrum antibodies may form the basis for antibody therapeutics to be locally delivered on the gastrointestinal tract.

Kanwar et al., in a DSS model of colitis, used oral delivery of bovine milk-derived iron-saturated lactoferrin (Fe-Blf), angiogenin, osteopontin colostrum whey protein, Modulen IBD, cis-9, trans-11 conjugated linoleic acid (CLA)-enriched milk fat and showed that each milk component attenuated experimental colitis but with a different effectiveness against specific disease parameters. So, they noticed a decrease in cytokine expression in mice fed with the treatment diets compared with DSS mice control group. The Fe-Blf, CLA-enriched milk fat, and diet rich in TGF inhibited intestinal angiogenesis, the CLA-enriched milk fat increased mouse body weight, reduced epithelium damage, and downregulated the expression of proinflammatory cytokines and nitrous oxide. Osteopontin (OPN) lowered the inflammatory score. Myeloperoxidase activity was lower in mice fed with the special diet, OPN, angiogenin, and Fe-Blf [61].

Filipescu et al., in a TNBS model of colitis in rats, noticed that mice pretreated daily with a suspension of bovine colostrum have significantly less intestinal damage and clinical signs of the colitis. They noticed a reduction in body weight loss and histological score of the treated animals as compared to colitis group. The expression levels of TLR4 IL-1β IL-8 and IL-10 were lower in mice receiving bovine colostrum. Interestingly enough, they found no significant changes in bacterial load after the induction of TNBS colitis in colostrum pre-treated mice [10].

In a recently published experimental study, Spalinger et al. using two models of colitis, namely DSS-colitis and T-cell transfer colitis, showed that oral administration of IMM-124E, a colostrum-based product containing anti-E.coli-LPS IgG, reduces intestinal inflammation. More specifically, they noticed a reduction in the serum LPS-binding protein and in flow cytometry, reduced numbers of effector T helper cells, and increased levels of regulatory T cells. They concluded that IMM-124E might represent a novel therapeutic strategy to induce or maintain remission in chronic colitis [62].

COLOSTRONONI is a new dietary supplement consisting of bovine colostrum and *Morinda citrifolia* fruit (Noni), the latter being a plant, native to the Indian Ocean, that produces a large number of phytochemicals. This dietary supplement might prevent intestinal inflammation and the development of chronic inflammatory disorders [63]. In an in vitro model of intestinal epithelium, COLOSTRONONI stimulated cell turnover and increased the gene expression of IL-8, two factors being fundamental for the establishment of mechanisms necessary to repair tissue damage [64]. COLOSTRONONI needs to be further explored in clinical trials concerning patients with inflammatory bowel diseases.

In conclusion, the available (mainly experimental) data showed promising results concerning the role of colostrum in IBD. Large clinical studies, in which colostrum should be administered either alone or with other factors, including TGF in an enteral nutrition formula, are needed.

## 4. Probiotics in Inflammatory Bowel Disease

A satisfactory volume of data, including metanalyses, concerning the role of various types of pro- and pre-biotics in patients with active IBD, especially UC, have been recently published, all of which described promising results. However, and despite the positive results achieved in the great majority of the published studies, a permanent echo in the conclusion part of almost all articles is repeated, claiming that “*…despite these promising results, more data derived from large double-blind studies are needed…*” Certainly, the probiotic bacteria used in many studies were not selected on the basis of their immune-modulating properties; rather on the basis of their availability and survival in the upper gastrointestinal tract. Moreover, in some studies, a rather insufficient dosage of probiotics was used. Despite these weaknesses, probiotics could play a rather important role in the treatment of patients with IBD in conjunction with the established medical treatment.

In this section, we shall summarize the data derived from published meta-analyses performed during the last two years (2018 and 2019), regarding the real value of probiotics in the treatment of IBD patients.

### 4.1. Probiotics and IBD Pathophysiology

It is well-established that gut microbiota composition and metabolism are correlated with the host immune system. Therefore, administration of pro- and pre-biotics could modify the gut microbiota of the patient, leading to a clinical improvement. As mentioned before, a number of in vitro, in vivo and clinical studies in IBD patients evaluating many probiotic formulations, (especially VSL#3), in active disease and in maintaining remission, have been published [65]. In patients with UC a clinical evidence of efficacy for some specific strains and especially for multi-strain preparations, certainly exists [66]. Knox et al. recently stressed that more encouraging results are emerging as the microbial composition of these probiotics have been optimized in an effort to include bacterial strains showing clear clinical benefit. On the other hand, it is widely accepted that fecal transplantation represents a promising microbiome-modulating treatment in refractory UC, indicating that novel therapies for IBD patients should include a microbiome-modulating approach with a personalized and multidimensional approach [67].

Probiotics exert their beneficial effect through multiple mechanisms, including stimulation of anti-inflammatory and inhibition of pro-inflammatory cytokines, restoration of the intestinal mucosal barrier, antagonistic action on many pathogens, and many others [68] (Table 3). Such mechanisms have been investigated in animal models.

### 4.2. Effectiveness of Probiotics in IBD: Results of the Published Meta-Analyses

There were six metanalyses published during the last two years [13,69,70,71,72,73], the results of which are summarized in Table 4.

The conclusions that we can draw from these metaanalyses could be the following: The administration of probiotics either alone or in combination with mesalazine could increase the rate of clinical remission of patients with UC. No significant differences in the rate of maintenance of remission between patients with UC either receiving probiotics or mesalazine were found, meaning that probiotics are at least equally effective as mesalazine in keeping patients with UC in remission. The VSL#3 formula achieved the most promising results. No differences in the rate and type of side-effects between the groups of patients receiving probiotics and patients receiving mesalazine, were noticed. At least two meta-analyses confirmed the assumption that probiotics act synergistically with mesalazine significantly increasing the therapeutic benefit. The effectiveness of probiotics in CD patients is weak. However, the combination of *Saccharomyces boulardii*, *Lactobacillus*, and VSL#3 showed a trend for efficiency.

Apart from these metaanalyses, a pilot study comparing probiotics with placebo in their overall oxidant ability and antioxidant response in patients with IBD has recently been published. Ballini et al. showed that the values of oxidative stress and anti-oxidant response observed in the group of patients receiving probiotics for three months were significantly improved, as compared with the group of patients receiving placebo. The authors concluded that the oral administration of specific probiotics could be efficacious and safe in patients with UC [74].

Again, in a very recently published study, Bjarnason et al. found that the administration of a multi-strain probiotic (Symprove™, Symprove, UK) in patients with IBD was associated with a decreased intestinal inflammation in patients with UC, but not in patients with CD [75]. In another study, concerning the efficacy of different probiotic strains, Alard et al. [76] have tested six strains concerning their ability to improve gut permeability, as well as act as anti-inflammatory agents using in vivo and in vitro models. They found that *Bifidobacterium bifidum* PI22 strain, while exhibiting significant protective capacities against acute colitis, was slightly efficacious in chronic colitis. On the other hand, *Bifidobacterium lactis* LA804 strain, although it showed weak efficacy in the acute model of colitis, exhibited a significantly protective action against chronic colitis. Moreover, *Lactobacillus helveticus* PI5, although it has not shown anti-inflammatory abilities in vitro, has demonstrated a strong epithelial barrier restorative activity, thus improving murine acute colitis. Finally, *Lactobacillus salivarius* LA307 significantly protected mice against both types of colitis. This study identified four strains having a high potential for the management of IBD.

In conclusion, from the available data it can be assumed that probiotics (especially the VSL#3) are useful agents for maintaining remissions in patients with pouchitis or active ulcerative colitis, and Bifidobacterium for treating patients with UC in remission [77]. When using probiotics in large doses per day in active disease, the gastroenterologist should bear in mind that this beneficial strategy might be even detrimental for the patient [78]. As previously mentioned, in order to achieve the most optimal results, future work should focus on specific combinations of probiotics that could target specific immune sites in the gut. Moreover, controlled trials with uniform criteria are necessary in order to clarify the efficacy of probiotics, and optimize the clinical results.

## 5. Enteral Formulas Supplemented with TGF Used in IBD Patients in Clinical Studies

During recent years, industry efforts, aiming to preserve the biological activity of some bioactive molecules in end products, resulted in the production of food containing TGF𝛽 [6,7,8]. Recently, more enteral formulas supplemented with TGFβ2 appeared in the market. One of them (Santactiv Digest powder) is a polymeric diet rich in TGFβ2 from colostrum, supplemented with probiotics (Bacillus coagulans) and *n*-3 fatty acids. The main nutritional values of this formula per 60 g powder are as follows: Energy: 300 kcal, lipids 13 g, carbohydrates 32 g, proteins 13 g, fiber 0.1 g vitamins A, C, E, B complex, D3, K1, trace elements, L-glutamine 2 g, colostrum 0.6 g and probiotics 5.0 mld. The 60 g of powder should be diluted in 190 mL of fresh water and consumed in about half an hour. Each meal offers 300 kcal. It is not suitable to be used as source of nutrition. The dosage recommended is two sachets of 60 g per day in the active phase of CD, and one sachet of 60 g per day as a nutritional support during the remission phase [79].

There is a considerable speculation concerning the mode of action of these formulas in IBD. It seems that in CD endogenous healing pathways mediated by TGFβ are inhibited because mucosal inflammatory cells express Smad7, the endogenous intracellular inhibitor of TGFα signaling [80]. It seems unlikely that enteral feeds containing TFGβ exert their therapeutic effect by means of direct anti-inflammatory effects, although TGFβ may promote mucosal healing in synergy with changes in mucosal bacterial populations as a result of the change in the diet. Antigen exclusion and changes in bacterial flora seem to be the most important [81]. An important anti-inflammatory effect of TGFβ is the promotion and generation of FOXP3-positive regulatory T cells in the intestinal compartment [82].

A relatively unanswered question is related to the ability of TGFβ to pass the whole digestive tract without degradation by the digestive enzymes. According to Beattle et al. [83], a survey of the TGFβ content of twenty milk-based preparations demonstrated that its presence depends on the source of milk protein and processing conditions. It seems that casein itself may inhibit the enzymatic degradation of TGFβ by the duodenal and enteric juice [84].

So far, very few studies concerning the role of a diet rich in TGFβ in improving experimental colitis, have been published. Oz et al. investigated the effect of an orally administered diet containing TGFβ2 on intestinal injury and immune responses in an IL-10 knockout mouse model of colitis. At the end of the 8th week mice fed with the diet rich in TGF-β2 significantly increased their body weight and hematocrite, and decreased the levels of serum amyloid A and TNFα, compared to the control diet group. Concerning histology, a lower score of severity was noticed in the TGFβ2 group as compared with the control group [85]. In an interesting study, treatment with TGFβ2 of rats with experimentally methotrexare-induced intestinal mucositis prevented mucosal-injury, enhanced p-ERK and β-catenin induced enterocyte proliferation, inhibited enterocyte apoptosis and improved intestinal recovery [86].

As far as it concerns the clinical efficacy of enteral nutrition rich in TGFβ in IBD patients, very few data are available. These studies are analyzed subsequently. Ferreira et al. assessed the effects of nutrition supplementation with and without TGFβ2 on inflammatory, endoscopic, histopathologic, and nutrition parameters in 38 patients with active CD. Patients were divided into three groups: group 1 (patients with nutrition orientation), group 2 (nutrition orientation and normal nutrition), and group 3 (nutrition orientation and nutritional supplement with TGFβ2) for 3 months. At the end of the study, the CDAI was reduced in groups 2 and 3, while in group 3 a reduction in C-Reactive Protein (CRP) levels and an improvement in histologic findings were observed. They concluded that, despite the improvement in nutrition and inflammatory patterns, only those patients receiving TGFβ2-enriched formula showed an improvement in histologic parameters and a reduction in CRP levels [87].

Beaupel et al. evaluated the effect of preoperative administration of exclusive TGFβ2 in decreasing postoperative complications after surgery for complicated ileocolonic CD (35 high vs. 21 low-risk patients). The result showed that the discontinuation of steroids in preoperative TGFβ2 group was feasible in 62.5%. No significant differences in the complication rates between the two groups were observed. Thus, the preoperative administration of TGFβ2 diet is feasible and beneficial in high-risk patients with complicated CD, as it could effectively reduce the postoperative morbidity, although the results should be confirmed in large randomized controlled trials [88].

Davanço et al. described the effect of a diet supplement containing whey proteins and TGFβ on the body composition of 42 consecutive patients with CD. They noticed that patients supplemented with this regime showed an increase in their lean body mass, as compared with the non-supplemented group. They concluded that whey protein intake supplemented with TGFβ can improve the Lean Body Mass, while concurrently reducing the fat percentage [89].

In another study, 29 adult patients with active CD received a special diet supplemented with TGFβ as an exclusive diet for 4 weeks at a dose of 50 g × 5/d. Clinical improvement was noticed in 69% of the patients. All nutritional parameters were improved. Patients stopped losing weight and the score of general well-being increased. No change of the situation or worsening was noticed in 9 patients (31%). The results suggested that this diet could be effective in inducing remission in a proportion of adult patients with mild CD [6]. Regarding the long-term efficacy of TGFβ rich diet in maintaining remission, Triantafillidis et al. compared the results of the administration of a diet rich in TGFβ with those of mesalazine in a group of patients with CD in remission. Patients were assigned to receive either two meals (2 × 50 g) of the special diet plus two regular meals per day or mesalazine (800mg three times a day) for six months. At the end of the trial, 69% of patients in the special diet arm continued to be in remission compared with 60% of those receiving mesalazine (no significant differences). The mean time from remission to relapse was 103 days versus 123 days, respectively [7]. However, when interpreting these results, one should bear in mind the current knowledge of the lack of efficacy of mesalazine in maintaining remission in Crohn’s disease patients [90]. A quite interesting finding of this study was the increase in the levels of HDL and the decrease in the levels of LDL in patients receiving the special diet. The role of dysfunctional HDL in cytokine induction and inflammation seems to be quite important, as HDL can modulate LDL oxidation and LDL-induced cytokine production and inflammation [91]. Dysfunctional HDL has been identified in animal models and humans with chronic inflammatory diseases. It seems that the anti-inflammatory properties of HDL may be at least as important as the levels of HDL-cholesterol. The pathophysiological consequences of the administration of a diet rich in TGFβ needs further exploration, as the levels of HDL and LDL before and after treatment of IBD patients could constitute a useful index of inflammatory activity.

## 6. Other Nutritional Compounds

### 6.1. Short Chain Fatty Acids (SCFAs)

SCFAs are fatty acids with less than six carbons and including five acids, namely acetate, formic, propionate, butyrate, and valeric acid. SCFAs are secondary metabolites produced through the fermentation of dietary substrates (proteins, peptides, resistant starches, and undigested fibers) by the gut microbioma. SCFAs represent the exclusive energy source for the intestinal epithelial cells. The production of these fatty acids is influenced by certain parameters, including host nutrition, and the existence and concentration of specific commensal bacteria [92]. For example, propionate and acetate are produced by *Firmicutes* and *Bacteroidetes* phylum respectively, through the lactate and succinate pathway [93]. In addition, the production of butyrate, acetate, and propionate, possess different ratios in different sites of the intestinal tract, as well as different physiological activities; e.g., propionate and acetate could be present in both large and small intestines, while butyrate is mainly present in cecum and left colon [94]. SCFAs may be absorbed by passive diffusion, although energetic uptake by the intestinal epithelial cells could also be succeeded via special substrate transporters [95].

SCFAs support and maintain the intestinal homeostasis and the gut barrier function having direct or indirect effects on intestinal epithelial cells, including proliferation, differentiation, and gene expression. SCFAs exhibit modulating effects on immune cells (e.g., T cells, especially Tregs, neutrophils, and macrophages). In fact, the SCFAs can affect cytokine production and migration, cytolytic activity, and epigenetic modulation. SCFAs may signal through cell surface G-protein coupled receptors to activate signaling cascades that control immune functions [96]. Butyrate, probably the most important SCFA, influences both the adaptive and the innate immunity, regulating the activation of regulatory T cells, and decreasing the activation of NF-ΚB. Butyrate, also, increases the mucus production and the rearrangement of tight junction proteins [97].

In patients with IBD gut SCFAs are reduced, although the results of the relevant studies could be characterized as conflicting. In a recently published systematic review and metaanalysis, Zhuang et al. found significant alterations of SCFAs in IBD patients; e.g., in patients with active and inactive ulcerative colitis inverse SCFA alterations [98]. The significance of these fatty acids in the pathogenesis of IBD is further supported by the fact according to which bacteria in mucosa and feces of patients with IBD producing SCFAs are reduced. IBD patients show reduced levels of dominant SCFAs-producing bacteria (like *Faecalibacterium prausnitzii* and *Roseburia intestinalis*) in intestinal mucosa and feces, and lower levels of SCFAs compared to healthy controls [99].

It is well-established that intestinal inflammatory responses are modulated by the gut microbiome. The importance of microbiota in controlling inflammation is shown when a bowel segment is excluded from the fecal stream (diversion colitis/pouchitis) [100]. IBD patients show dysbiosis and loss of microbiome diversity, and the associated alterations in SCFA levels might be restored by fecal microbiota transplantation obtained from healthy donors [101].

In conclusion, SCFAs represent important dietary metabolites that might be useful in the treatment of patients with IBD. The use of prebiotics and probiotics and fiber-rich diets may improve the clinical condition of patients with IBD by improving the levels of SCFAs and especially butyrate, in the bowel lumen.

### 6.2. Flavonoids

Flavonoid compounds are a large family of hydroxylated polyphenolic molecules abundant in plants, including vegetables and fruits, which are the major dietary sources of these compounds for humans, along with wine and tea. Flavonoids are becoming very popular as they have many health-promoting and disease-preventive effects. Most interest has been directed towards the antioxidant activity of flavonoids, evidencing a remarkable free-radical scavenging capacity. Flavonoids have many other biological properties, including anti-inflammatory, antiviral, anticancer, and neuroprotective activities [102]. Key mechanisms relate to their anti-inflammatory and antioxidant properties, as well as their ability to modulate gut microbiota, have been recently described [103]. *Go to*:

Several studies asserted the anti-IBD effects of *Citrus* fruits and their flavonoids, thus providing evidence favoring their role in the prevention and treatment of IBD. In a prospective study of preillness diet in newly diagnosed patients with Crohn’s disease, Octoratou et al. found that foods decreasing the risk of acquiring Crohn’s disease on logistic regression analysis, were citrus fruits [104]. Other studies considering the use of plants containing flavonoids in patients with IBD have shown that these plants may actually induce clinical response and, to a lesser degree, clinical remission [105]. A pilot clinical study performed in patients with ulcerative colitis refractory to 5-ASA and/or azathioprine has revealed beneficial results [106].

In conclusion, it could be argued that flavonoids could potentially be used as an effective treatment in patients with IBD. Their anti-inflammatory effects and the mechanisms of action have been confirmed in experimental models being similar to those of the drugs currently used in patients with IBD, including biologic agents [107]. On the other hand, fruits and vegetables represent a very safe and without risk source of flavonoids making them a cheap and widely available therapeutic modality. Further clinical studies should be performed in order to elucidate the efficacy of various flavonoids in patients with IBD.

### 6.3. Bioactive Peptides

It is well-known that the digestibility of dietary proteins is higher than 90%. However, a certain amount of luminal proteins may escape digestion in the small intestine and be transferred into the large intestine. It has been estimated that between 6 and 18 g of protein reach the colon serving as a nitrogenous substrate for the microbiota activity in the large intestine. The origin of protein (animal or plant) influences the nature and quantity of amino acids delivered in the large bowel lumen. Increased protein intake could result in gut bacterial dysbiosis by altering the microbiota composition and the mucosal immune system. Bacterial metabolites derived from amino acids, such as H_2_S and ammonia, could have a positive or negative impact on the intestinal epithelium with some of them being detrimental and some others beneficial.

Epidemiological evidence indicates that the consumption of high animal protein is associated with an increased risk of IBD [108]. Food components could modulate DNA methylation and a number of epigenetic mechanisms, thus predisposing to IBD development [109].

A lot of published data confirmed that peptides originating from different food sources have antiinflammatory, antioxidant and immunomodulatory properties [110]. For instance, amino acids derived from dietary proteins display beneficial effects helping the macromolecule synthesis in the inflamed bowel mucosa. However, an excessive amount of dietary proteins may result in an increased production of deleterious bacterial metabolites causing the inhibition of colonic epithelial cell proliferation, and increasing intestinal permeability [111]. The impact of the high protein diet on the gut microbiota and colonic epithelium might be amplified in patients with a compromised barrier function, such as patients with IBD.

Bioactive peptides that could modulate the intestinal cytokine milieu in normal people might be beneficial for keeping patients with IBD in remission and preventing relapses, although they are not able to restore the altered mucosal cytokine profile in patients with active IBD [78]. Therefore, dietary peptides and amino acids could be useful as alternative treatments in IBD [112]. The exact role of dietary peptides and amino acids should be further investigated in order to develop patient-tailored diets for IBD patients.

## 7. Conclusions

The available data indicate that EN enriched in TGFβ, apart from restoring the impaired nutritional status, could also modulate intestinal immune responses, thus positively affecting the inflammatory bowel processes. This type of EN could be characterized as a supporting and/or primary therapy aiming to induce and/or maintain remission. These formulas enriched with colostrum and probiotics of a special amount and consistency might be proved of benefit in patients with IBD in the near future. In our opinion, it is important not to underestimate the role of nutrition as supportive care in patients with CD. Further well-designed large trials with enteral formulas containing TGFβ, colostrum, probiotics and other nutritional compounds are necessary in order to improve the results of EN.

## Figures and Tables

**Table 1 nutrients-12-01048-t001:** Trophic and bioactive factors in colostrum and milk.

• Antioxidants
*α-Carotene* *β-Carotene* *Lycopene* *Retinol* *α-tocopherol* *γ-tocoferol*
• **Nonpeptide trophic factors**
*Glutamine* *Polyamines* *Nucleotides*
• **Hormones** • **Cytokines** • **Growth factors**
*Epidermal Growth Factor* (*EGF*) *Transforming Growth Factor-a* (*TGFa*) *Transforming Growth Factor-β family* (*TGFβ*) *Platelet-derived Growth Factor* *Vascular Endothelial Growth Factor* (*VEGF*) *Growth Hormone and its Releasing Factor* *Hepatocyte Growth Factor* (*HGF*) *Neuronal Growth Factors* *Insulin-Like Growth Factor* (*IGF*) *Superfamily*

**Table 2 nutrients-12-01048-t002:** Studies on the role of colostrum in experimental colitis.

Reference	Chemical Colitis	Experimental Design	Results	Conclusion
Bodammer et al. 2011 [56]	DSS- colitis	Colostrum vs. BSA vs. water for 2 week	Improvement of clinical and histological severity. Redistribution of immune-regulatory, peripheral and splenic gd TCR+ and CD11b+Gr1+ cells.	Improvement of symptoms and inflammation.
Kailash et al. 2013 [57]	DSS- and TNBS- Induced colitis	Isolation of AVX-470 and AVX-470m from colostrum of dairy cows immunized with TNF vs. infliximab	Orally administered AVX-470m reduced disease severity. AVX-470 has in vitro activity comparable to that of infliximab.	Oral administration of this antibody is effective in treating mouse models of IBD.
Kanwar et al. 2016 [61]	DSS-colitis	Oral delivery of bovine milk-derived Fe-bLF, angiogenin osteopontin, colostrum, whey protein, Modulen IBD cis-9,trans-11 conjugated linoleic acid (CLA)-enriched milk fat	Decrease in cytokine expression. Fe-bLF, CLA-enriched milk fat, and CLA-enriched milk fat reduced epithelium damage, and down-regulated the expression of proinflammatory cytokines. Myeloperoxidase activity was lower in mice fed Modulen IBD, OPN, angiogenin, and Fe-bLF.	Each milk component attenuated experimental colitis but with different effectiveness against specific disease parameters.
Filipescu et al. 2018 [10]	TNBS- Induced colitis	Mice received a daily suspension of bovine colostrum or saline solution for 21 days before TNBS colitis.	Reduction in BW loss and histological score compared to CN. Lower expression of TLR4 IL-1β IL-8 and IL-10	Pre-treatment with bovine colostrum reduces the intestinal damage and signs of colitis.
Spalinger et al. 2019 [62]	DSS- colitis and T-cell transfer colitis:	IMM-124E a colostrum-based product containing anti-E.coli-LPS IgG.	Amelioration of DSS colitis and T cell transfer colitis. Reduction in infiltrating immune cells. Reduced numbers of effector T helper cells, increased levels of regulatory T cells.	Oral IMM-124E reduces intestinal inflammation.

IBD: Inflammatory Bowel Disease, CN: normal controls, BW: Body Weight.

**Table 3 nutrients-12-01048-t003:** Main mechanisms of action of probiotics (modified from Triantafillidis et al. [68]).

Antimicrobial effect	Decreased colonization and invasion by pathogenic organisms Modification of pH Production of inhibitory substances Block of adhesion sites Competition for essential nutrients Degradation of toxin receptor
Restoration of gut integrity	Restoration of intestinal permeability Up-regulation of mucosal barrier function with up-regulation of tight junction molecules
Modification of the host immune response	Reduction in proinflammatory cytokine content on plasma and lymphocytes Decrease in the colonic concentration of IL-6, TNFα NF-kB and p65 Reduction in leukocyte recruitment. Decrease in colonic MPO activity Expansion of mucosal regulatory cells

**Table 4 nutrients-12-01048-t004:** Systematic reviews and metaanalyses concerning the role of probiotics in IBD.

Reference	No of RCTs/ pts	Disease	Type of Study	Probiotic Used	Results	Conclusion
Astó et al. 2019 [69]	18 studies 1419 patients	UC active	Probiotics vs. placebo vs. active treatment	Bifido- bacteria	No significant differences for placebo or mesalazine-controlled studies	Bifidobacteria: Promising for active UC
Peng et al. 2019 [70]	27 studies 1942 patients	UC active	Probiotics with 5-ASA vs. 5-ASA vs. Sulfasala- zine		Remission rate: higher in the group of probiotics plus mesalazine, vs. mesalazine Probiotics combined with mesalazine increased the remission rate in active UC.	Probiotics combined with 5-ASA increase the remission rate in active UC.
Chen et al. 2019 [71]	60 studies 4954 patients	UC active	Bifid probiotic plus 5-ASA vs. 5-ASA alone	bifid triple viable probiotic (BTV)	BTV plus mesalazine improved the remission rate and reduced the relapse rate. Levels of cytokines were reduced and levels of IL-10, CD3+, CD4+, were increased.	Combination treatment of BTV with mesalazine improved active UC.
Jia et al. 2018 [72]	10 studies 1049 patients	IBD	Probiotics vs. Placebo	*E coli* Nissle 1917 and VSL#3	No differences on remission, relapse, and complication rate between probiotics and placebo group. VSL#3: higher remission rate and lower relapse rate.	*E coli* Nissle 1917 and VSL#3: alternative therapy for IBD.
Ganji-Arjenaki et al. 2018 [13]	9 and 18 studies for CD and UC respect-tively	UC and CD including pediatric population		VSL#3 *Lactoba-cillus* Combination of *Saccharo-myces boulardii*, *Lactobaci-* *llus*, and VSL#3 In children, combination of *Lactobacillus* and VSL#3	Analysis of 9 trials: Probiotics had not significant effect on CD. Analysis of 3 trials in children: Significant improvement. Analysis of 18 trials: UC: significant effects. VSL#3: significant effect *Lactobacillus*: significant effect in UC. Combination of *Saccharomyces boulardii*, *Lactobacillus*, and VSL#3 in CD: A trend for efficiency In children: combination of *Lactobacillus* with VSL#3 had significant effect.	Probiotics are beneficial in IBD and especially in patients with UC, if they are administered in combination.
Derwa et al. 2017 [73]	22	UC or CD	Probiotics vs. placebo vs. 5-ASA	VSL#3	No benefit of probiotics in active UC Trials of VSL#3: clear benefit. Probiotics equivalent to 5-ASA in preventing UC relapse. No benefit of probiotics in active CD, and in preventing relapses after surgery.	Probiotics are equivalent to 5-ASA in preventing relapse of quiescent UC. Efficacy in CD uncertain.

UC: ulcerative colitis, CD: Crohn’s disease.

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
