# Peer review of "Enteral Nutrition Supplemented with Transforming Growth Factor-β, Colostrum, Probiotics, and Other Nutritional Compounds in the Treatment of Patients with Inflammatory Bowel Disease"

_nutrients, 2020, doi:10.3390/nu12041048_

Round 1

Reviewer 1 Report

  • “Enteral nutrition (EN) can reduce Crohn's disease (CD) activity and maintain remission in both children and adults [1].”

Cite a study regarding children too

  • “Kailash et al investigated two polyclonal antibodies AVX-470 and AVX-470m specific for human TNF isolated from colostrum of dairy cows that had been previously immunized with TNF, in two models of experimental colitis (DSS and TNBS-induced colitis) [52]. These orally administered antibodies significantly reduced the severity of colitis in a manner comparable with that of oral prednisolone. The authors concluded that AVX-470 polyclonal anti-TNF antibody has in vitro activity comparable to that of infliximab.”

Generally, antibodies are given intravenously or subcutaneously because are big proteins (the gastric passage would destroy it): explain why these two polyclonal antibodies could be used orally

  • Explain the meaning of “colostro noni”

  • “Regarding the long-term efficacy of TGF-β rich diet in maintaining remission, Triantafillidis et al compared the results of the administration of a diet rich in TGF-β with those of mesalazine in a group of patients with CD in remission. Patients were assigned to receive either two meals (2×50 g) of the special diet plus two regular meals per day or mesalazine (800 mg three times a day) for six months. At the end of trial, 69% of patients in the special diet arm continued to be in remission compared with 60% of those receiving mesalazine (no significant differences). The mean time from remission to relapse was 103 days versus 123 days respectively”

Cite the lack of efficacy of mesalazine in maintaining the remission in Crohn’s disease (Gomollón F et al. 3rd European Evidence-based Consensus on the Diagnosis and Management of Crohn's Disease 2016: Part 1: Diagnosis and Medical Management. J Crohns Colitis. 2017 Jan;11(1):3-25.)

Reviewer 2 Report

The review describes the role of enteral nutrition in inflammatory bowel disease (IBD), in particular, that of derived components concerning the fractions of TGF-beta, colostrum and probiotics. Enteral nutrition is an interesting topic in the current research of IBD; though noticeable variability between studies is still found and evidence about nutritional factors is still not consistent. Therefore, the proposed review adds some novelty, is timely, and may be of potential interest for IBD scientific community. However, some aspects described in the following lines must be improved in the manuscript:

* Major comments.

- The concepts introduced about the implication of each compound (TGF-beta, colostrum and probiotics) in the pathophysiology of IBD need to be further improved. They are described in a confusing manner for the reader, they need to be more concise and clear.

This is particularly evident in the section regarding colostrum. Terms in Table 2 make no sense, with the corresponding writing in lines 175-205. The terms “Action” and “Pathophysiology” is Table 2 are confusing. Why authors indicated in line 177 “The pathophysiological actions of colostrum are shown in table 2 and analyzed subsequently [43]”? Reference [43] is included or discussed?

- Tables should include concise concepts following a schematic manner. Moreover, the content of Tables (particularly the long sentences) should not be repeated in the writing.

This is particularly evident in Table 3 and Table 7.

- Concepts on pathophysiology are confusing. However, considering the main scope of the journal “Nutrients”, authors may consider to at least define introductory concepts on other nutritional compounds; other than TGF-beta, colostrum and probiotics, known to modulate the inflammatory processes, particularly IBD, and the microbiota. These may include recent reviews on bioactive compounds such as flavonoids (World J Gastroenterol. 2017, 23, 5097-5114), short chain fatty acids (Curr Pharm Des. 2018, 24, 4154-4166) and proteins and bioactive peptides (Trends Food Sci Technol. 2019, 88, 194-206), among others.

- Probiotics (line 284) “An alteration in the gut microbiota is a key factor needed to be present for the appearance of IBD”. It is currently unknown whether microbiota dysbiosis is cause/consequence in the inflammatory process. This need to be further verified.

- Probiotics. They may be even detrimental in case of active disease. From previous publication in “Nutrients” (2019, 11, 2605) “(…) gut commensals and peptides thereof may be beneficial as novel nutraceutical compounds, which may help to maintain local homeostasis in health or patients in remission, but they might be detrimental as therapeutic agents in active IBD.”. Authors may need to consider this perspective in the manuscript. Moreover, the reference Am J Gastroenterol. 2018, 113, 1125-1136. may be of particular interest for this section.

- Table 6. Please, consider if it is relevant for the review to include as a Table all the nutritional values of a specific enteral formula?

- It is again confusing the inclusion of 5.1 Mechanisms of action and 5.2 Experimental studies, on Section 5. presumably dedicated to enteral formulas for IBD patients (clinical studies).

- From my perspective, some references are not suitable or accurate to support the affirmations proposed. Please, update some references and carefully check those more relevant to the findings of this field, while it may be more rigorous not to consider some individual findings not well supported in the literature. This is needed to avoid the misinterpretation of the current proved findings and the addition of controversial results.

* Minor comments:

- Abbreviation for TGF is sometimes used for TGF-beta, some others for the superfamily, or for the isoforms, sometimes in plural, some others in singular. Please, verify the writing and typing. There is an unclear use of abbreviations in several points of the manuscript.

- Line 61. The use of “superficial zones” referring to the gut epithelium is not scientifically appropriate.

- Line 111. Regulation of inflammatory processes, particularly, in the gut. This need to be further described, in a non-ambiguous manner.

- Line 113. Please verify the use of italics.

- Lines 140-142. Please verify the spaces between numbers. Carefully check the entire manuscript for typing errors.

- Lines 285. Microflora. The term microbiota may be more appropriate in this context.

  • In Tables including information about a published study, include the corresponding number of the references “[]”.

  • There is a general poor language use, and it is recommend that authors have their paper language edited.

Round 2

Reviewer 2 Report

The authors have successfully raised most of my concerns and improved the quality of the review.